# Heteroscedastic Heatmap Regression for Reliable Pectoral Muscle Segmentation in Mammography

**Paul Zech**[1,2]  🆔        P.ZECH@SIEMENS-HEALTHINEERS.COM
**Christian Hümmer**[1]        CHRISTIAN.HUEMMER@SIEMENS-HEALTHINEERS.COM
**Benjamin El-Zein**[1,2]        BENJAMIN.EL-ZEIN@SIEMENS-HEALTHINEERS.COM
**Christopher Syben**[1]        CHRISTOPHER.SYBEN@SIEMENS-HEALTHINEERS.COM
**Ludwig Ritschl**[1]        LUDWIG.RITSCHL@SIEMENS-HEALTHINEERS.COM
**Steffen Kappler**[1]        STEFFEN.KAPPLER@SIEMENS-HEALTHINEERS.COM
**Sebastian Stober**[2]        STOBER@OVGU.DE

[1] *Siemens Healthineers AG, Forchheim, Germany*

[2] *Otto-von-Guericke University, Magdeburg, Germany*

**Editors:** Accepted for publication at MIDL 2026

## Abstract

Breast cancer remains a leading cause of mortality worldwide, making accurate mammography screening essential for early detection. An important preprocessing step in mammography is the accurate segmentation of the pectoral muscle, as it affects downstream tasks such as breast density estimation or automated exposure control. Existing automated segmentation methods, both traditional and deep learning-based, often lack reliable confidence measures, which becomes especially problematic in the presence of occlusions or visually confounding structures such as skin folds or other muscle fibers. To address this limitation, we propose a probabilistic framework that combines heatmap-based boundary regression with heteroscedastic uncertainty estimation to capture input-dependent variability. Our approach not only predicts the pectoral muscle boundary but also quantifies the associated uncertainty. While mainly producing unimodal predictions, the probabilistic heatmaps reveal multimodal patterns for confounding structures, further enhancing transparency in challenging cases. We demonstrate that our method provides robust and transparent means to achieve accurate segmentation while producing meaningful uncertainty estimates.

**Keywords:** Pectoral muscle segmentation, heteroscedastic regression, aleatoric uncertainty

## 1. Introduction

Breast cancer remains one of the most prominent cancer types, especially in young and middle-aged women (Siegel et al., 2016; Ren et al., 2022). To improve early detection, the World Health Organization recommends regular mammography screening in this population (World Health Organization, 2014). One standard projection for breast tissue characterization is the mediolateral oblique (MLO) view in mammograms. Besides breast tissue, MLO projections also capture the pectoral muscle (PM), which typically appears as a bright and dense region in the upper corner of the image. Accurate segmentation of the PM constitutes an essential preprocessing step in the analysis of mammography images. For instance, breast positioning control systems use the PM as a key anatomical landmark (Brahim et al., 2022). Moreover, PM segmentation is important for automated patient-specific calibration

of the radiation exposure, which is computed on a previously acquired low-dose preshot. Accurate exclusion of the PM is essential to prevent the dose from being calibrated to the dense PM tissue, which would otherwise result in elevated radiation and overexposure. To automate the PM segmentation task, many different traditional and deep learning-based algorithms have been developed. However, the proposed solutions provide no or insufficient measures of confidence for their predictions (Rampun et al., 2019; Ma et al., 2019). This is particularly problematic in PM segmentation, as the PM may be obscured by dense glandular tissue or confused with structurally similar skin folds and other muscle fibers. In such ambiguous cases, uncertainty modeling is essential to identify potentially unreliable local segmentation results, especially in high-noise environments such as low-dose preshots.

To address these challenges, we propose a novel framework for PM segmentation that explicitly accounts for multiple sources of input noise and ambiguities. We reformulate the task as contour regression, modeling the contour positions as probability distributions that capture both the expected contour position and its associated uncertainty. We predict these distributions jointly by means of a single probabilistic heatmap using an image-to-image (I2I) architecture. Our method builds on previous work (Hümmer et al., 2024) with substantial technical improvements, which we validate through a comprehensive experimental evaluation.

## 2. Related work

**PM segmentation:** Early developments leverage prior knowledge about the shape of the PM boundary by detecting a straight line (Karssemeijer, 1998) or fitting active contours to the PM boundary (Ferrari et al., 2004; Wang et al., 2010). Other traditional algorithms can be categorized into line detection, intensity-based, wavelet-based, and statistical techniques as summarized in Ganesan et al. (2013). However, these methods rely on extensive pre- and postprocessing as well as feature engineering.

Recent advances in deep learning overcome these limitations by learning hierarchical representations directly from data using deep neural networks. Some approaches address the task through pixel-wise classification of the PM tissue. Architectures such as U-Net (Ronneberger et al., 2015) and its variants have been widely adopted for this purpose (Ma et al., 2019; Liu et al., 2020), with extensions that incorporate adversarial training to improve robustness and anatomical plausibility (Guo et al., 2020). While pixel-wise classification can model highly complex shapes, this flexibility is not required for PM segmentation in MLO images: since the muscle occupies a well-defined position in the upper image corner, its area is fully determined by the outer contour alone. Therefore, contour-based methods can focus on the clinically challenging part of delineating the PM boundary, which may overlap with dense breast tissue. These methods are based on edge detection or boundary-aware strategies, such as VGG16-based contour detection (Soleimani and Michailovich, 2020), U-Net adaptations for boundary extraction (Angelone et al., 2025), and Holistically-Nested Edge Detection (Rampun et al., 2019). By restricting the prediction task to the PM boundary, the solution space is reduced, simplifying the learning problem while still fully defining the muscle region (Angelone et al., 2025). Building on this idea, Hümmer et al. (2024) exploited anatomical prior knowledge by showing that the PM boundary admits a unique functional mapping from image rows to column indices. This reformulation enables segmentation to

be performed as a row-wise column-index (CI) regression, inherently encouraging continuity and structural consistency of the predicted PM boundary. The authors were able to show that this approach notably outperforms a standard pixel-wise classification baseline in terms of segmentation performance, parameter efficiency, and inference time. However, despite their high accuracy, the lack of a robust measure of uncertainty remains a key limitation of these solutions.

**Uncertainty quantification:** Following Kendall and Gal (2017), uncertainty arises from two sources: (1) epistemic uncertainty, due to limited training data and uncertain model parameters, and (2) aleatoric uncertainty, caused by noise or ambiguity in the input. Aleatoric uncertainty can further be classified as homoscedastic (constant across inputs) or heteroscedastic (input-dependent). However, in medical imaging, most methods estimate the overall predictive uncertainty rather than explicitly modeling either one (Lambert et al., 2024). Common strategies involve modeling a predictive distribution through Monte-Carlo (MC) dropout with multiple stochastic forward passes at test time (Gal and Ghahramani, 2016; Jungo et al., 2018), or through ensembles of differently initialized models (Mehrtash et al., 2020). In PM segmentation, similar strategies have been applied using MC dropout (Klanecek et al., 2023) or deep ensembles, either from model snapshots along the training trajectory (Tang et al., 2025) or from models trained on different data distributions (Hümmer et al., 2024). However, these methods do not model input-dependent, heteroscedastic uncertainty to capture the aforementioned input ambiguities. A straightforward approach to model this uncertainty is to use the inter-rater variability as ground truth for the uncertainty in a supervised setting as done in Cetindag et al. (2022). A more scalable approach is to implicitly learn heteroscedastic uncertainty from the data itself. The underlying idea is to predict the mean and variance of the predictive distribution, which is learned by maximizing the Log-Likelihood (LL) within a heteroscedastic framework (Lambert et al., 2024). This is usually achieved by adding the variance as a separate output to the mean predictions and has been successfully applied to segmentation (Graham et al., 2020) and regression tasks (Seitzer et al., 2022). Nevertheless, to the best of our knowledge, heteroscedastic uncertainty modeling has not been applied to PM segmentation yet.

**Heteroscedastic heatmap regression:** The detection of a contour is closely related to coordinate regression, as a contour can be represented by a finite set of points. A common strategy for coordinate regression is heatmap-based regression, in which coordinate locations are encoded as Gaussian distributions within spatial heatmaps. In its simplest form, the network is trained to regress these target Gaussians centered at the ground truth coordinates using a heatmap matching objective (Zhang et al., 2020). This approach has also been applied in the context of heteroscedastic uncertainty modeling. For instance, Thaler et al. (2021) demonstrated that the learned heatmaps can be interpreted as pseudo-probability distributions that can be used to quantify uncertainty. However, their approach does not explicitly model heteroscedasticity during training, but instead fits a Gaussian model to the learned heatmaps during inference. Most other methods that model heteroscedastic uncertainty do not represent uncertainty as a full probability distribution over the heatmap. Instead, they predict parameterized Gaussians through mean and variance (Seitzer et al., 2022; Shukla et al., 2024), thereby imposing a strong unimodal Gaussian assumption. However, this assumption is not necessarily valid in PM segmentation, where confounding structures such as skin folds can lead to multimodal uncertainty patterns. To

our knowledge, only Kumar et al. (2020) attempted to model full heteroscedastic distributions directly in the heatmap in the context of landmark detection, but reported instabilities due to the limited spatial resolution of current heatmap-based frameworks.

**Our contribution:** In this study, we advance the idea of Hümmer et al. (2024), who modeled PM segmentation as row-wise CI regression of the PM boundary. To allow uncertainty in the contour predictions, we replace the discrete CI vector with a probabilistic heatmap where each row represents a full probability distribution over possible contour positions. We demonstrate that the probabilistic heatmaps can be learned using an I2I network trained with a heteroscedastic loss function and show that robust uncertainty estimates can be derived from the predicted heatmaps' distributions. We further show that the prediction of full row-wise distributions enables the identification of uncertain cases where the underlying assumption of unimodality on the errors is violated. Last, we perform a rigorous evaluation in terms of method configuration, segmentation performance, and ability to quantify uncertainty. A preliminary version of this work was presented in Zech et al. (2025); this paper presents the full, extended study.

## 3. Methods

An overview of our method is depicted in Figure 1. Input images are first processed by a U-Net to produce a spatial heatmap, which is then converted into a probabilistic heatmap using a row-wise softmax operation. The contour prediction and its associated uncertainty are subsequently extracted as mean and variance, which are used for training with a heteroscedastic loss function. An additional regularizer ensures that the model produces valid probability distributions. All steps are detailed below.

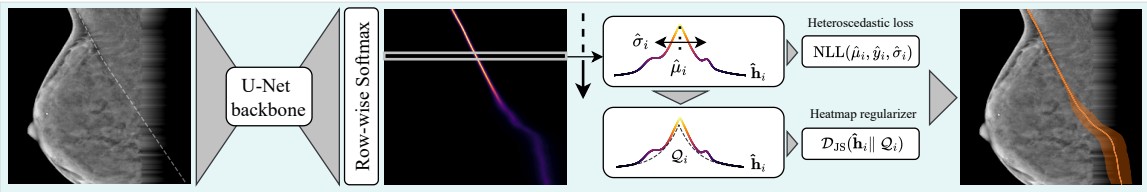

Figure 1: Overview of our method, with label-extrapolated input (left), predictive heatmap (center) and boundary prediction with uncertainty band (right).

**Probabilistic heatmap regression:** First, the input image $\mathbf{I} \in \mathbb{R}^{H \times W}$ is processed by an I2I network to generate a spatial heatmap $h$. For this task, we employ a U-Net as described in the original publication (Ronneberger et al., 2015), as it is widely adopted and well established in medical image analysis. To obtain the probabilistic heatmap, we convert each row into a pseudo-probability distribution by applying a row-wise softmax operation

$$\hat{h}_{i,j} = \mathrm{Softmax}(h_{i,j}) = \frac{\exp(h_{i,j})}{\sum_{w=0}^{W-1} \exp(h_{i,w})}, \tag{1}$$

where $\hat{h}_{i,j}$ is the softmax-activated heatmap $h$ at row $i \in \{0, \ldots, H-1\}$ and column $j \in \{0, \ldots, W-1\}$, and index $w$ runs over all columns of row $i$. In this probabilistic

heatmap, we define the PM contour prediction as the row-wise mean of the learned probability distributions. Accordingly, we define the uncertainty as a row-wise variance over the predicted boundary positions. The row-wise mean and variance are computed as first- and second-order moments directly from the probabilistic heatmap as

$$\hat{\mu}_i = \sum_{j=0}^{W-1} j \cdot \hat{h}_{i,j}, \qquad \hat{\sigma}_i^2 = \sum_{j=0}^{W-1} \hat{h}_{i,j} \cdot (j - \hat{\mu}_i)^2 \tag{2}$$

with mean $\hat{\mu}_i$ as PM boundary prediction and variance $\hat{\sigma}_i^2$ as a measure of the uncertainty for row $i$. The mean computation in Equation (2) corresponds to the soft-argmax operation (Luvizon et al., 2019), which has been proven effective in different scenarios, as it leverages the spatial generalization of fully-convolutional networks (FCNs) (Nibali et al., 2018).

To model heteroscedastic uncertainty in the probabilistic heatmap, we train the network by minimizing the negative log-likelihood (NLL) derived from both Laplace ($\mathcal{L}$) and Normal ($\mathcal{N}$) distributions, both well known for this purpose (Kumar et al., 2020), with

$$\text{NLL}_{\mathcal{N}} := \frac{1}{H} \sum_{i=0}^{H-1} \left( \frac{(y_i - \hat{\mu}_i)^2}{2\hat{\sigma}_i^2} + \frac{1}{2} \log(2\pi\hat{\sigma}_i^2) \right) \quad \text{and} \tag{3}$$

$$\text{NLL}_{\mathcal{L}} := \frac{1}{H} \sum_{i=0}^{H-1} \left( \frac{|y_i - \hat{\mu}_i|}{\hat{b}_i} + \log(2\hat{b}_i) \right), \quad \text{where } \hat{b}_i = \sqrt{\frac{\hat{\sigma}_i^2}{2}}. \tag{4}$$

Here, $y_i$ refers to the ground truth contour position at row $i$. In this setting, the shape of the heatmaps' distributions is only weakly supervised, since infinitely many different distributions can yield the same first- and second-order moments, which may lead to unstable training behavior. To address this, we extend the loss function with a regularization term that enforces a Gaussian or Laplacian shape on the predicted distributions, thereby stabilizing soft-argmax-based training as demonstrated by Nibali et al. (2018). Unlike static variance regularization, our approach employs a variable regularization where both the mean and variance are controlled by the heteroscedastic loss (see Figure 1). In more detail, we define the regularizer as

$$\mathcal{L}_{\text{reg}} = \frac{1}{H} \sum_{i=0}^{H-1} \mathcal{D}_{\text{JS}}(\hat{\mathbf{h}}_i \| \mathcal{Q}_i) \quad \text{with} \quad \mathcal{Q}_i = \begin{cases} \mathcal{N}(\hat{\mu}_i, \hat{\sigma}_i^2) & \text{(Gaussian)} \\ \mathcal{L}(\hat{\mu}_i, \hat{b}_i) & \text{(Laplace),} \end{cases} \tag{5}$$

where $\mathcal{D}_{\text{JS}}(\hat{\mathbf{h}}_i \| \mathcal{Q}_i)$ denotes the Jensen-Shannon divergence between $\hat{\mathbf{h}}_i$ defined as the softmax-activated heatmap in row $i$, and a template distribution $\mathcal{Q}_i$, which is constructed from the predicted heatmaps' statistics in Equation (2). The regularizer, scaled by a constant factor $\lambda$ to control its strength, is added to the heteroscedastic loss to encourage distributions that align with the probabilistic assumptions of the loss function.

**Detecting multimodality:** As previously discussed, many uncertainty modeling approaches assume unimodal error distributions (Sec. 2). However, this assumption might not hold for all anatomical situations. For instance, confounding structures can induce bi- or

multimodal patterns in the predictive distributions, as multiple structures may represent plausible anatomical interpretations of the PM. Since our method predicts full probability distributions, it enables identifying such cases where the underlying unimodal assumption is violated. This is achieved by classifying whether there exists at least one row in the predicted heatmap that contains more than one peak. In this setting, the regularizing term serves as a soft constraint that controls the trade-off between enforcing row-wise unimodal predictions for accurate training and allowing the model to express genuine multimodality in the predictions.

**Label extrapolation:** Representing the PM target contour as a CI vector requires defining a contour position for every image row. In the lower part of the image, where the muscle is no longer visible (see Figure 1), Hümmer et al. (2024) set the target CI to the image boundary. This introduces artificial flat segments, which are unproblematic for standard regression approaches that focus solely on point estimates. However, when modeling uncertainty, this leads to systematic overconfidence near the image edges. To avoid this, we pad the image by one-quarter of its width and linearly extrapolate the label vector as illustrated in Figure 1. This preserves a smooth muscle shape beyond the visible image area and provides a more robust basis for uncertainty quantification. Note that label extrapolation is used only for training with heteroscedastic losses to prevent overconfidence. For the baselines, we create the target CI vectors as described in Hümmer et al. (2024).

## 4. Experiments and results

**Dataset and labels:** To evaluate the proposed approach, we extracted 2,847 unprocessed MLO-view mammograms from the MBTST dataset (Dahlblom et al., 2019). Segmentation labels were provided by clinical experts as binary masks. To obtain contour labels, we converted the segmentation masks into a CI target vector as follows: for each image row, the corresponding entry of the CI vector was extracted as the first non-zero pixel in the respective binary mask. The dataset was split into training, validation, and test sets using a ratio of 75%/15%/10% with a uniform distribution of breast densities. This dataset split was kept consistent across all experiments.

**Image processing:** The images were processed by cropping to the region of interest, resizing to a resolution of $256 \times 256$ and subsequently Z-normalized.

**Augmentation:** For data augmentation, we employed the image processing pipeline of Eckert et al. (2024). The pipeline processes raw mammograms using a linear workflow comprising a Neg-Log transform, background segmentation, Laplacian-pyramid-based frequency band manipulation, and window leveling. All configurable parameters are mapped to three normalized values in $[0, 1]$, representing realistic imaging styles. During training, we sampled these parameters uniformly, while during inference we kept them fixed to $[1.0, 0.5, 0.5]$[1]. For detailed information about the image processing pipeline, refer to Eckert et al. (2024).

**Training:** As the optimizer, we used AdamW with weight decay $10^{-2}$ and an initial learning rate of $10^{-5}$, which was reduced by a factor of 0.1 when the loss plateaued for more than 10 epochs. For all experiments, we present the average metrics across three independent training runs. Throughout the experiments, uncertainty is quantified by computing the row-wise standard deviation $\hat{\sigma}_i$ from the variance defined in Equation (2) and subsequently

---

1. The first parameter controls window leveling, where 1.0 corresponds to the largest possible window.

averaging it over all rows, with the resulting mean-standard-deviation (MSTD) reported as the uncertainty metric.

## 4.1. Heatmap regression configuration

To determine an effective heteroscedastic training setup, we systematically compare different loss formulations and regularization strengths within our heatmap regression framework. Specifically, we evaluate the heteroscedastic loss functions, $\text{NLL}_\mathcal{N}$ in Equation (3) and $\text{NLL}_\mathcal{L}$ in Equation (4), against two regression baseline losses, mean-absolute-error (MAE) and mean-squared-error (MSE), for different regularization strengths $\lambda$. Since MAE and MSE provide no variance supervision, the fixed variance $\sigma_i^2 = 10$ was empirically chosen as a target for the template distributions $\mathcal{Q}_i$ in the regularizer (Equation (5)). For evaluation, we choose MAE and root-mean-squared-error (RMSE) as standard regression metrics, while the LL is used to assess the quality of the learned predictive distributions. All metrics are computed solely within the muscle region and summarized in Table 1.

Table 1: Evaluation of models trained with two heteroscedastic losses ($\text{NLL}_\mathcal{L}$, $\text{NLL}_\mathcal{N}$) and two baseline losses (MAE, MSE) across three regularization strengths $\lambda$. Reported metrics are MAE, RMSE, and LL, given as mean $\pm$ standard deviation over 3 runs.

| Loss | $\lambda$ | MAE $\downarrow$ | RMSE $\downarrow$ | LL $\uparrow$ |
|------|-----------|------------------|-------------------|---------------|
| MAE | 0 | $1.97 \pm 0.04$ | $2.46 \pm 0.05$ | $-3.55 \pm 0.19$ |
| | 10 | $1.94 \pm 0.02$ | $2.40 \pm 0.03$ | $-2.55 \pm 0.07$ |
| | 100 | $1.90 \pm 0.04$ | $2.35 \pm 0.04$ | $-2.38 \pm 0.03$ |
| MSE | 0 | $1.95 \pm 0.07$ | $2.43 \pm 0.06$ | $-3.85 \pm 0.21$ |
| | 10 | $1.93 \pm 0.02$ | $2.41 \pm 0.04$ | $-3.43 \pm 0.36$ |
| | 100 | $1.90 \pm 0.01$ | $2.38 \pm 0.03$ | $-2.83 \pm 0.19$ |
| $\text{NLL}_\mathcal{L}$ | 0 | $2.01 \pm 0.00$ | $2.58 \pm 0.02$ | $-2.30 \pm 0.02$ |
| | 10 | $1.92 \pm 0.04$ | $2.42 \pm 0.05$ | $-2.28 \pm 0.02$ |
| | 100 | $1.88 \pm 0.04$ | $2.34 \pm 0.05$ | $-2.26 \pm 0.02$ |
| $\text{NLL}_\mathcal{N}$ | 0 | $2.03 \pm 0.03$ | $2.59 \pm 0.10$ | $-2.31 \pm 0.01$ |
| | 10 | $1.98 \pm 0.01$ | $2.51 \pm 0.01$ | $-2.32 \pm 0.08$ |
| | 100 | $1.94 \pm 0.09$ | $2.42 \pm 0.11$ | $-2.30 \pm 0.07$ |

The results show that the introduction of the regularization term leads to a slight performance improvement, as both MAE and RMSE decrease with increasing regularization strength across all used loss functions. A similar trend is observed for the LL values, which also increase with stronger regularization. Furthermore, both heteroscedastic loss formulations achieve MAE and RMSE values comparable to the non-heteroscedastic loss functions. At the same time, the heteroscedastic models show high LL values across all regularization strengths. Overall, the lowest errors and highest LLs are achieved by the model trained with $\text{NLL}_\mathcal{L}$. These findings suggest that incorporating heteroscedastic uncertainty model-

ing and label extrapolation does not compromise predictive performance while it allows for learning stable predictive distributions across different regularization strengths. Further, it is demonstrated that the regularization term stabilizes training notably, as evidenced by consistent improvement across all metrics and loss functions for higher regularization strengths. Among the evaluated configurations, the model trained with $\text{NLL}_\mathcal{L}$ achieves the lowest errors and highest LLs, indicating that the Laplace distribution is better suited to model the underlying heteroscedastic uncertainty. It is therefore selected as the best configuration and used in all subsequent experiments, and referred to as $\text{NLL}_\mathcal{L}$-model.

### 4.2. Performance comparison

To verify that our approach does not compromise segmentation performance, we compare it against a pixel-wise classification baseline. For this, we choose a classical U-Net (Ronneberger et al., 2015), trained to predict a binary segmentation mask using a binary cross-entropy loss. We assess the segmentation performance in terms of Dice across different network sizes in Figure 2.

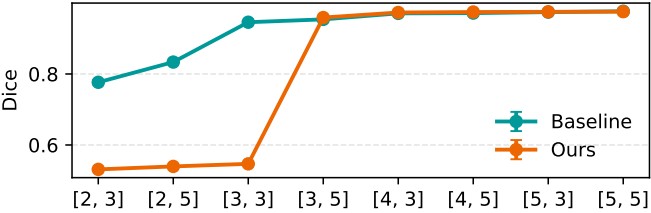

Figure 2: Segmentation performance (Dice) of our method (orange) against a U-Net baseline (blue) across network sizes [depth, filters], where depth is the number of U-Net stages and filters the number of first-layer filters (doubled each stage).

For very small networks, the segmentation performance of our method is noticeably lower than that of the baseline. This indicates that the model capacity is insufficient to handle the increased complexity of the task, which involves modeling both the contour and the associated uncertainty, as well as interpolating in regions where the muscle is not visible. In contrast, for medium and large network configurations, performance is on par with the baseline, while the model simultaneously provides uncertainty estimates with high LLs as highlighted in Table 2.

Table 2: LL achieved by our method for different network configurations (filters \ depth), reported as mean ± standard deviation over three runs.

| LL (filters \ depth) | 2 | 3 | 4 | 5 |
|---|---|---|---|---|
| 3 | $-5.10 \pm 0.01$ | $-4.95 \pm 0.02$ | $-2.27 \pm 0.01$ | $-2.25 \pm 0.01$ |
| 5 | $-5.06 \pm 0.00$ | $-2.51 \pm 0.02$ | $-2.28 \pm 0.04$ | $-2.26 \pm 0.02$ |

### 4.3. Uncertainty quantification

In this section, we examine the method's ability to quantify uncertainty. We evaluate the model's response to (1) inherent ambiguities in the dataset, such as occlusions or confounders, and (2) artificially introduced unseen noise corrupting the input images. In practice, we assess how well the uncertainty correlates with model errors by means of mean residuals (MAE) and the predicted MSTD. The experiments are conducted on the $\text{NLL}_{\mathcal{L}}$-model trained with $\lambda = 100$ (Sec. 4.1) and the results are shown in Figure 3.

**(1) Dataset-intrinsic uncertainty:** To analyze heteroscedastic uncertainty originating from the dataset itself, we employ a similar strategy to Kumar et al. (2020): We perform the model inference on the test set and collect all row-wise residual errors and predicted heatmap standard deviations for every image. We then sort these tuples by standard deviation and group them into bins of size $N_{\text{bin}} = 50$. Within each bin, we compute the average residual error (MAE) and the MSTD; each bin therefore corresponds to a single point in Figure 3(a).

**(2) Uncertainty under unseen noise:** To evaluate whether the model's uncertainty estimates generalize to unseen input noise, we add noise to the unprocessed input images during inference. The noise is modeled by artificially reducing the photon counts and adding X-ray-typical Poisson noise to the input images using the procedure described by Eckert et al. (2019)[2]. To this end, we linearly reduce the effective dose from 100% to 25% in 50 steps while producing 3 realizations with different seeds for each dose level. Finally, the model generates predictions for all noisy image realizations and the results are aggregated over all images. Hence, each dot in Figure 3(b) represents MAE and MSTD, aggregated over the whole test set at the respective noise level.

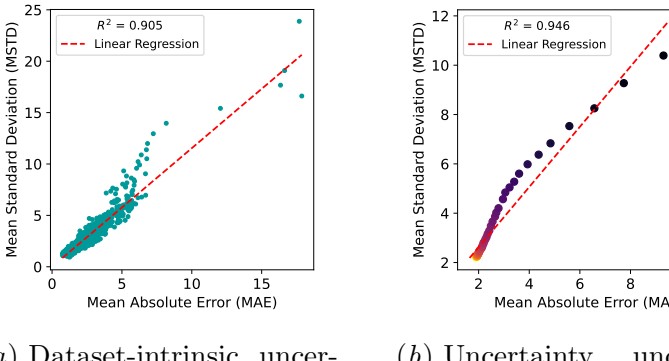

(a) Dataset-intrinsic uncertainty.

(b) Uncertainty under unseen noise.

Figure 3: Correlation between MAE and MSTD for inherent noise in the dataset 3(a) and unseen noise distorting the input 3(b) for the $\text{NLL}_{\mathcal{L}}$-model ($\lambda = 100$).

For the uncertainty within the test set in Figure 3(a), the results show a strong linear correlation between the MAE and MSTD, with the MSTD consistently matching or slightly exceeding the MAE. A similar behavior can be observed for the model's response to pre-

---

2. Only the Anscombe transformation is omitted as it is not required for noise simulation.

viously unseen noise in Figure 3($b$). Artificially reducing the dose leads to a continuous increase in model error to which the model responds with a steady increase in the MSTD of the predicted heatmap. These results indicate that the standard deviations computed from the model's predictive distributions are highly predictive for model error for both inherent heteroscedastic noise within the dataset and previously unseen noise. It should be noted that this finding holds as an aggregated behavior when averaging over $N_{\text{bin}}$ for Figure 3($a$) or the whole test set for Figure 3($b$).

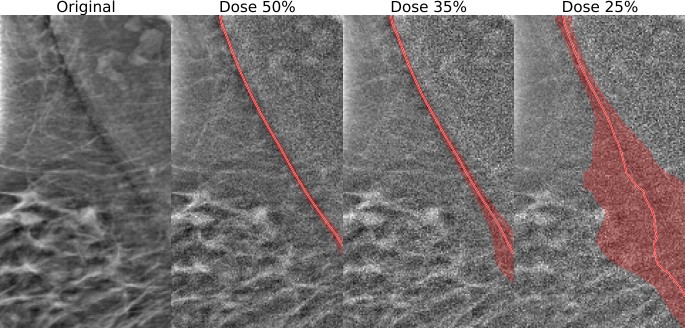

Figure 4: Predictions of the NLL$_{\mathcal{L}}$-model ($\lambda = 100$) for an example case from the test set across different noise levels. The row-wise mean $\hat{\mu}_i$ is depicted as a thick red line and the corresponding standard deviation $\hat{\sigma}_i$ as a light red area around the mean.

To complement the quantitative analysis, we examine the model's behavior at the image level using an exemplary image under varying noise levels shown in Figure 4. Up to a 50% dose reduction, the model produces accurate predictions with low uncertainty along the entire contour. When further reducing the dose to 35%, the model's uncertainty starts to increase locally in the lower part where the muscle is originally slightly occluded, while still producing a stable contour prediction. At 25% dose, the contour prediction degrades notably in the lower part of the muscle, but at the same time the model increases uncertainty significantly as an indicator for prediction failure. The observations suggest that the model is able to adjust uncertainty locally in areas of reduced visibility, while maintaining stable and confident predictions in regions where the muscle remains clearly visible.

### 4.4. Detecting multimodality in the predictive distributions

This section evaluates the method's capability to reveal multimodal patterns in the learned distributions. Further, we evaluate how the regularization term affects the number of detected cases that exhibit multimodal distributions in the predicted heatmaps. To this end, we compare the proportion of detected cases that exhibit multimodal distributions across different regularization strengths for the NLL$_{\mathcal{L}}$-model and evaluate the performance in terms of MAE and LL within the classified subgroups of uni- and multimodal cases. The results are summarized in Table 3.

The results show that for stronger regularization, more cases are classified as unimodal. At the same time, MAE and LL worsen notably within the multimodal class while only

Table 3: Subgroup analysis of the $\text{NLL}_{\mathcal{L}}$-model based on predicted heatmaps: unimodal vs. multimodal. MAE and LL are shown for each subgroup across different $\lambda$, with proportions of each class. Metrics are mean $\pm$ standard deviation over 3 runs.

| Metric | Modality | $\lambda = 0.0$ | $\lambda = 10.0$ | $\lambda = 100.0$ |
|---|---|---|---|---|
| MAE $\downarrow$ | Multimodal | $3.49 \pm 0.11$ | $3.82 \pm 0.03$ | $6.08 \pm 1.32$ |
|  | Unimodal | $1.72 \pm 0.05$ | $1.78 \pm 0.05$ | $1.85 \pm 0.03$ |
| LL $\uparrow$ | Multimodal | $-2.77 \pm 0.06$ | $-2.84 \pm 0.03$ | $-3.57 \pm 0.66$ |
|  | Unimodal | $-2.21 \pm 0.03$ | $-2.24 \pm 0.03$ | $-2.25 \pm 0.02$ |
| **Proportion** [%] | Multimodal | 16.15 | 7.05 | 0.91 |
|  | Unimodal | 83.85 | 92.95 | 99.09 |

marginally deteriorating for the unimodal class. This suggests that the regularization term substantially stabilizes the predicted heatmaps towards unimodal distributions, forcing the model to suppress smaller confounding structures. As a result, only large confounders remain, which are responsible for the larger prediction errors. To further support this, we analyze two qualitative examples of high uncertainty from both classes for $\lambda = 0$ and $\lambda = 100$, depicted in Figure 5. For the unimodal case in Figure 5(a), it is observed that the

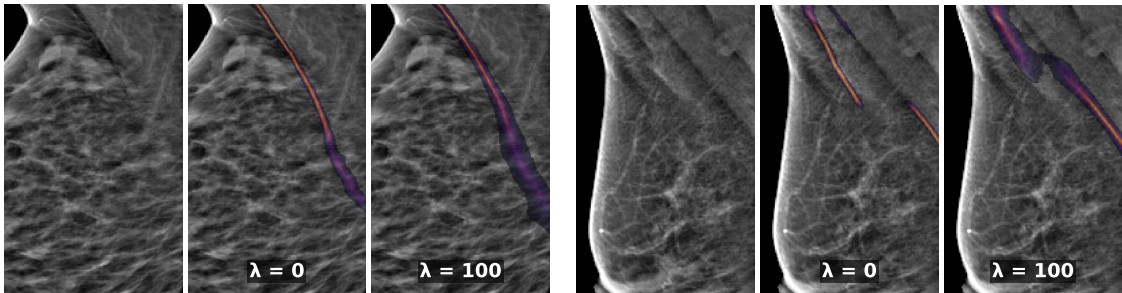

($a$) Unimodal case: muscle occluded by breast tissue.  ($b$) Multimodal case: confounding structures.

Figure 5: Qualitative examples illustrating (a) unimodal and (b) multimodal predictive distributions for the $\text{NLL}_{\mathcal{L}}$-model for two regularization strengths $\lambda$.

muscle shows a clear edge in the upper part while being occluded by breast tissue in the lower part of the muscle, to which both models react with increased uncertainty. For the multimodal case in Figure 5(b), there are multiple confounding anatomical structures in the image, leading to multimodal predictive distributions in the heatmaps. This aligns with our initial assumption that in the confounding case the underlying estimation problem becomes inherently multimodal, whereas in the occluded scenario, the potential contour location can be adequately represented by a unimodal distribution. The results indicate that the model can capture multimodal patterns in the distributions, with the sensitivity adjustable via the regularization term. Finally, the results show that the regularizer notably stabilizes

the heatmaps towards smooth unimodal distributions for larger regularization strengths, especially for the confounding structure in Figure 5(b).

### 4.5. Limitations

In this section, we discuss the limitations regarding the evaluation of our method. First, all results are averaged over three independent model runs, chosen due to resource constraints. While this allows certain insights into the stability of the results with respect to different model initializations, three runs are not sufficient to compute valid statistics. However, we report the mean and standard deviation across runs to illustrate the consistency of the observed trends.

Further, in Section 4.3, the uncertainty estimates are aggregated over bins of size $N_{\text{bin}} = 50$ rows and over the entire dataset, respectively. While this aggregation captures global uncertainty trends, it does not allow conclusions about the model's local or spatially resolved uncertainty behavior, which is discussed qualitatively in this study.

Finally, the study focuses on one representative model architecture and one dataset. While this allows a focused evaluation, it may limit the generalizability of the findings across different architectures and datasets.

## 5. Conclusion and Outlook

In this work, we present a novel method for modeling input-dependent uncertainty in PM segmentation using a heatmap-based heteroscedastic regression framework. We show that robust mean and variance estimates can be derived from learned probabilistic heatmaps to jointly model the PM boundary and the associated predictive uncertainty. Furthermore, by representing uncertainty directly in probabilistic heatmaps, the method provides richer information than approaches that output mean and variance as isolated numerical values, as it allows detecting inherent multimodality in the predictive distributions and controlling this behavior through a dedicated regularizer. At the same time, we show that our method does not compromise segmentation performance as we achieve on-par performance with a binary segmentation baseline. Last, we show that the model's uncertainty estimates correlate with model error in a global trend and demonstrate that the model reacts appropriately to previously unseen noise, increasing its predicted uncertainty when reduced visibility of the PM leads to erroneous predictions.

Although the current framework mainly models unimodal distributions, it establishes a strong foundation for future research on extending the approach to explicitly model multimodal predictive distributions. Moreover, a systematic evaluation against multi-reader annotations would provide valuable insights into the method's ability to capture localized uncertainty. Such an analysis, combined with validation on larger and more diverse datasets, would be essential to ensure robustness in clinical practice.

**Disclaimer:** The methods presented in this paper are not commercially available and their future availability cannot be guaranteed.

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
