# OpenReview forum: "Heteroscedastic Heatmap Regression for Reliable Pectoral Muscle Segmentation in Mammography"
_MIDL.io/2026/Conference — MIDL 2026 Poster_

### Official Review · Reviewer_9G98 · 2026-01-08

**Confidence:** 3
**Preliminary Rating:** 4
**Final Rating:** 5

**Summary:**

The paper presents theory and results from predicting and modeling uncertainty of pectoral muscle (PM) detection in mammography using a regularized heteroscedastic loss. Several quantitiative and qualitative results from PM detection, downstream segmentation, and uncertainty quantification are presented.

**Strengths:**

- Well written paper with substantial background and clean math
- Regularizing clearly shows improvement over
- Results support the usefulness of new heteroscedastic loss function for quantifying uncertainty
- Mean prediction and quantified uncertainty are well-aligned upon visual inspection and with noise imputation.

**Weaknesses:**

- The paper does not clearly describe the model architecture implemented and trained using the loss functions. Reproducibility is difficult due to lack of code / clear description
- Results reported from single model and single dataset. Unclear if hte method can generalize well.
- While there is substantial amount of new results compared to their previous work (Zeck et al 2025), the methodological novelty is unclear. This is only a concern as the authors suggest in Introduction and Methods that this work is different from the previous work methodologically as well. But most of the math overlaps.
- Mean and standard deviation of the results are reported only with 3 random initialization. Typically >5 samples are required to assess variability. A stronger result would have been if the dataset splits were cross-validated instead of fixed train-test splits. In either case, p-value would be helpful which is currently not reported
- Some missing description and clarifications which can be fixed. (See detailed comments)

**Detailed Comments:**

- Support the line "However, the proposed solutions provide no or insufficient measures of confidence." with citation. Could be the same ones as the ones appearing in Related work.
- Clarify why the NLL would compute model uncertainty. Compare against previous work used NLL to compute data uncertainty [ Kendall and Gal 2017]. Clarify the intuition.

    - Kendall, Alex, and Yarin Gal. "What uncertainties do we need in bayesian deep learning for computer vision?." Advances in neural information processing systems 30 (2017).

- Clarify if the mean-standard-deviation is computed row-wise as per the methodology in the sentence "Throughout the experiments, the model uncertainty is quantified as mean-standard-deviation (MSTD) of the predicted heatmaps distributions."
- Clarify the term "label extrapolation". (This could be a very application specific jargon that I am unaware of.) Why do some methods need label extrapolation (e.g., MAE)
- What is the prediction performance metric reported in Table 2? Why does it differ from table 1?
- Fig 2 should ideally show examples for \lambda = 0 as well to judge the improvement provided by regularization qualitatively.

**Justification Of Final Rating:**

The authors have clearly and succinctly clarified all my concerns. The paper now clearly shows the benefits of having uncertainty-aware regression for mammogram segmentation. Several experiments show the use-cases and ablations. New clarifications also enable reproducibility.

**Justification Of The Preliminary Rating:**

The method is very interesting and the presented results are substantially new and support the usefulness of presented method. Certain clarifications regarding novelty and connection to existing literature are required.

**Questions To Address In The Rebuttal:**

- Clearly describe the improvements from the previous pubvlication from the authors. Clarify the methodologival novelty if any. Explain why the results changed. ( could be because previous publication used ~8000 sample dataset and current publication used ~2000 samples ?)
- Describe model architecture implemented
- Report p-value / indicate statistically significant improvement in Tables. If possible, cross-validate the experimental setup
- Provide clarifications to comments in detailed comments around NLL, MSTD, label extrapolation, table 2, and fig 2.

---

> ### Author Response · Authors · 2026-01-23
> **Rebuttal response to reviewer 9G98**
>
> We thank the reviewer for the comprehensive review and the thoughtful feedback provided. We respond to the reviewer’s comments below.
>
> **"Clearly describe the improvements from the previous publication from the authors. Clarify the methodological novelty if any. Explain why the results changed."**
>
> We agree that the contribution relative to prior work was unclear in the initial manuscript. This study should not be understood as building on our previous **short paper** (Zech et al., 2025). The short paper served to outline the initial concept. The current study presents the fully developed version of that initial concept. More importantly, this work builds on the approach of Huemmer et al. (2024) by modeling the PM in a contour regression setting and introduces heatmap-based uncertainty modeling as a core contribution. To clarify this, we added a dedicated contributions section in Sec. 2 and revised the Introduction in Sec. 1.
>
> Nevertheless, we take the opportunity to outline changes to the initial concept that have affect on the results:
> - We use a different loss formulation: standard NLL in contrast to $\beta$-NLL
> - We use only **unprocessed** images which allows the use of the augmentation pipeline and enables the experimental setup that simulates Poisson noise (Sec.4.3)
>
> ---
>
> **“Describe model architecture implemented”**
>
> We use a standard U-Net, implemented according to the original publication. We added this in Sec.3.
>
> ---
>
> **“Report p-value / indicate statistically significant improvement in Tables. If possible, cross-validate the experimental setup”**
>
> We thank the reviewer for this suggestion. While we agree that averaging over only three runs is not sufficient to assess variability, we report the mean and variance to give an intuition about consistency in the results. Conducting a larger number of runs was not feasible given the number of experiments and shared resource constraints.
>
> We deliberately chose not to report inferential tests (e.g., p-values) because with only three runs, it is not possible to reliably estimate the underlying distribution or variance. Any p-value computed under these conditions could be misleading.
>
> Regarding cross-validation, this approach is most relevant in highly limited data scenarios. This is not the case here (> 2800 samples) and we believe that a separated and fixed test set - used consistently across all experiments - provides a more valid comparison.
>
> For transparency, we report the concern on the limited number of repetitions in a new dedicated limitations section (Sec.4.5).
>
> ---
>
> **“Support the line "However, the proposed solutions provide no or insufficient measures of confidence." with citation. Could be the same ones as the ones appearing in Related work.”**
>
> We added two citations of PM segmentation methods from the related work Section (Sec.2).
>
> ---
>
> **“Clarify why the NLL would compute model uncertainty. Compare against previous work used NLL to compute data uncertainty [Kendall and Gal 2017]. Clarify the intuition.”**
>
> This is a similar concern to reviewer xy9D. We explicitly only model for heteroscedastic data uncertainty in accordance with the definition in Kendall and Gal (2017). This concern most probably originates from a misleading use of the construction “Model uncertainty” in the initial manuscript. To avoid this confusion, we removed all occurrences in the revised manuscript.
>
> ---
>
> **“Clarify if the mean-standard-deviation is computed row-wise as per the methodology in the sentence: Throughout the experiments, the model uncertainty is quantified as mean-standard-deviation (MSTD) of the predicted heatmaps distributions.”**
>
> The MSTD is computed row-wise in accordance with the methodology. We clarified this in the revised manuscript in Sec.4.
>
> ---
>
> **“Clarify the term "label extrapolation". [...] Why do some methods need label extrapolation (e.g., MAE)”**
>
> The label extrapolation solves an application specific issue in the lower part of the images where the PM is not visible anymore that leads to overconfidence in uncertainty models. Consequently, it brings no advantage for the MAE-baseline and may even compromise its performance. We improved the description of the label extrapolation under a dedicated header in Sec.3.
>
> ---
>
> **“What is the prediction performance metric reported in Table 2? Why does it differ from table 1?”**
>
> We use the same metrics as in the previous experiment with MAE and LL (except RMSE). We agree that this was not clear in the initial manuscript, especially because the Table is transposed compared to Tab.1. For clarification we restructured both tables and revisited the captions. Note that Table 2 has been moved to a dedicated Section 4.4 and is now Table 3.
>
> ---
>
> **“Fig 2 should ideally show examples for \lambda = 0 as well to judge the improvement provided by regularization qualitatively.”**
>
> We agree, we added the images for \lambda=0 for both illustrated examples and revised the corresponding discussion in Sec.4.4.

---

### Official Review · Reviewer_xy9D · 2026-01-08

**Confidence:** 5
**Preliminary Rating:** 4
**Final Rating:** 5

**Summary:**

This paper proposes a heatmap regression method for segmenting pectoral muscles in mammography images and quantifying heteroscedastic data uncertainty. The problem addressed in the paper is meaningful and clinically relevant. The authors perform relevant experiments to demonstrate that their predictive uncertainty can capture data uncertainty. The paper is well structured (well organized). However, the methodological part of the paper deserves clarification: (1) to help the readers fully understand the technical developments proposed by the authors; (2) to be able to judge the rigor, quality, and integrity of the work; and (3) to guarantee the reproducibility of the paper.

**Strengths:**

The paper is well structured/organized and adequately addresses prior work. The problem the authors try to address in this paper is meaningful and clinically relevant. Uncertainty quantification of AI models in medical imaging is a cutting-edge research field that can have a significant clinical and industrial impact.

**Weaknesses:**

The writing of the paper can be improved, notably the methodological part of the paper. This part deserves clarification: (1) to help the readers fully understand the technical developments proposed by the authors; (2) to be able to judge the rigor, quality, and integrity of the work; and (3) to guarantee the reproducibility of the paper. The use of unnecessary pseudo-technical jargon could be diminished to improve the readability and fluidity of the paper. Additionally, the authors did not sufficiently justify their technical choices.

**Detailed Comments:**

Introduction:
The authors wrote “Moreover, it is important in automated exposure control, where a low dose image is acquired for breast density quantification and subsequent full-dose estimation”. What do you mean by subsequent full-dose estimation? Are you referring to the treatment (e.g., external beam radiotherapy)? Please clarify.

Related work:
The authors described two distinct approaches for segmenting pectoral muscles: pixel-wise classification and direct delineation of the contours. Provide the advantages and limitations of both approaches. Why did you develop a direct contour delineation method instead of a pixel-wise classification method? Justify your choices.

The paragraph titled “uncertainty-aware coordinate regression” is unclear. Please rewrite.

Method:
The method is mainly focused on the row direction. What is the performance of the method when the column direction is considered? Does mixing both directions improve the segmentation and uncertainty prediction?

Please describe precisely how the heatmaps were obtained. What loss function was used to train the U-Net? How did you build a ground truth to train the U-Net?

In the method workflow, the authors preliminarily obtained a contour prediction µ and a related uncertainty σ. Why is the heteroscedastic loss step needed? Please clarify.
Which model parameters does the heteroscedastic loss help to optimize?
Is the heteroscedastic loss used to train the whole method (including the U-Net)?

In equation (2), what is the meaning of k?

It is unclear how the heatmap regularization works and its purpose. Please clarify better.

Is it the uncertainty that is heteroscedastic or the regression? Please clarify and be consistent in the use of this word.

Experiments and results:
Which data augmentation was used to train the method? Provide details
Make a clear distinction between the preprocessing step of the images and the data augmentation steps.

The “Heatmap regression configuration” and “uncertainty quantification” sections are not clear. Please rewrite.

Table 1 is very unclear and confusing. It is difficult to fully understand the significance of the rows and columns and what the authors want to analyze or compare. Simplicity is usually better; please simplify this table.

The authors demonstrated that their predictive uncertainty can capture data uncertainty. It looks like this predictive uncertainty also contains model uncertainty. Can the proposed method isolate data and model uncertainties?

Conclusion and outlook:
Please indicate the limitations of the paper. Can the proposed method be used in clinical practice? What currently limits the use of this method for clinical practice?

**Justification Of Final Rating:**

I want to thank the author for their consideration of my comments, rigors, and professionalism. They replied precisely to all of my questions and drastically improved the clarity and readability of the paper.

I have a final request for the authors:
Could you clarify in the paper if the U-Net was pretrained, fine-tuned, or frozen during the training of your method? Or if the U-Net was end-to-end trained using their heteroscedastic loss? And what are the weights (networks or variables) updated by their heteroscedastic loss during training (it is not clear what is updated in an end-to-end fashion via their loss during training)?

**Justification Of The Preliminary Rating:**

The topic of the paper is clinically meaningful and within the scope of the conference. However, more works have to be done to ensure the readability, reproducibility, and scientific rigor of the paper.

**Questions To Address In The Rebuttal:**

I mainly would like that the authors take the time to improve the clarity, fluidity, and readability of the paper (notably the methodological and evaluation parts) and address the following concerns:

Introduction:
The authors wrote “Moreover, it is important in automated exposure control, where a low dose image is acquired for breast density quantification and subsequent full-dose estimation”. What do you mean by subsequent full-dose estimation? Are you referring to the treatment (e.g., external beam radiotherapy)? Please clarify.

Related work:
The authors described two distinct approaches for segmenting pectoral muscles: pixel-wise classification and direct delineation of the contours. Provide the advantages and limitations of both approaches. Why did you develop a direct contour delineation method instead of a pixel-wise classification method? Justify your choices.

The paragraph titled “uncertainty-aware coordinate regression” is unclear. Please rewrite.

Method:
The method is mainly focused on the row direction. What is the performance of the method when the column direction is considered? Does mixing both directions improve the segmentation and uncertainty prediction?

Please describe precisely how the heatmaps were obtained. What loss function was used to train the U-Net? How did you build a ground truth to train the U-Net?

In the method workflow, the authors preliminarily obtained a contour prediction µ and a related uncertainty σ. Why is the heteroscedastic loss step needed? Please clarify.
Which model parameters does the heteroscedastic loss help to optimize?
Is the heteroscedastic loss used to train the whole method (including the U-Net)?

In equation (2), what is the meaning of k?

It is unclear how the heatmap regularization works and its purpose. Please clarify better.

Is it the uncertainty that is heteroscedastic or the regression? Please clarify and be consistent in the use of this word.

Experiments and results:
Which data augmentation was used to train the method? Provide details
Make a clear distinction between the preprocessing step of the images and the data augmentation steps.

The “Heatmap regression configuration” and “uncertainty quantification” sections are not clear. Please rewrite.

Table 1 is very unclear and confusing. It is difficult to fully understand the significance of the rows and columns and what the authors want to analyze or compare. Simplicity is usually better; please simplify this table.

The authors demonstrated that their predictive uncertainty can capture data uncertainty. It looks like this predictive uncertainty also contains model uncertainty. Can the proposed method isolate data and model uncertainties?

Conclusion and outlook:
Please indicate the limitations of the paper. Can the proposed method be used in clinical practice? What currently limits the use of this method for clinical practice?

---

> ### Author Response · Authors · 2026-01-23
> **Rebuttal response to reviewer xy9D**
>
> We thank the reviewer for the detailed review and appreciate the time and effort invested in providing feedback. Questions are shortened and answers are kept short due to space limitations:
>
> **“What do you mean by subsequent full-dose estimation?”**
>
> We refer to the standard imaging workflow in mammography screening, in which an initial low-dose preshot is acquired to automatically estimate the appropriate exposure settings for the subsequent full-dose acquisition. We provided more details in Sec.1
>
> **“Why did you develop a direct contour delineation method instead of a pixel-wise classification method?”**
>
> The direct delineation method incorporates anatomical prior knowledge and was shown in Huemmer et al. (2024) to outperform pixel-wise classification baselines. We have added a dedicated contributions section and expanded the related work (Sec.2)
>
> **“The paragraph titled “uncertainty-aware coordinate regression” is unclear”**
>
> We agree that the original section was unclear and have addressed this as follows:
> - The section’s scope now focuses on heteroscedastic heatmap regression
> - An introductory sentence clarifies the relation to contour detection
> - The section content has been rewritten for improved readability
> - We added a broader perspective in the Introduction (Sec.1)
>
> **“Does mixing both directions improve the segmentation and uncertainty prediction?”**
>
> We appreciate this sharp observation. Unfortunately, extending the method to the column direction requires a unique (injective) mapping between the main axis and the contour position. This condition is naturally satisfied in the row direction because of anatomical reasons but does not always hold in the column direction. We added details to Sec.2 when describing the approach of Huemmer et al. (2024).
>
> **“Please describe precisely how the heatmaps were obtained. What loss function? “**
> **“Why is the heteroscedastic loss step needed? […] Is the heteroscedastic loss used to train the whole method?”**
> **“It is unclear how the heatmap regularization works and its purpose”**
>
> We believe that these questions arise from an unclear explanation in the methodological section. The input images are first processed by a U-Net to produce a spatial heatmap $h$. A row-wise Softmax is applied to obtain a probabilistic heatmap $\hat{h}$. From $\hat{h}$, the row-wise mean $\mu$ and variance $\sigma^2$ are computed according to Eq.(2), where $\mu$ models the PM contour and $\sigma^2$ the associated uncertainty. The entire method is trained end-to-end using the heteroscedastic losses defined in Eq.(3)+(4), which take $\mu$ and $\sigma$ as inputs. The regularization term in Eq.(5) is added to the loss functions to improve supervision of the heatmaps and encourage the underlying probability distributions (Gaussian/Laplace) of Eq.(3)+(4).
>
> We revised Sec.3 by adding an overview at the beginning, including the row-wise Softmax in Fig.1, and restructuring the text to describe the method in chronological order
>
> **“How did you build a ground truth to train the U-Net?”**
>
> We agree that the ground truth generation was not described in sufficient details and extended Sec.4 accordingly.
>
> **“In Eq.(2), what is the meaning of k?”**
>
> $k$ refers to the index the runs over all columns in row $i$ within the sum in the denominator. We added this to the manuscript.
>
> **“Is it the uncertainty that is heteroscedastic or the regression?”**
>
> We model heteroscedastic uncertainty (i.e., uncertainty that varies across inputs) by using a heteroscedastic regression model that predicts both a mean and an input-dependent variance. This terminology follows recent literature in [1,2], and the manuscript has been revised accordingly.
>
> **“Which data augmentation was used to train the method?”**
>
> We revised Sec.4 to distinguish between image preprocessing and data augmentation (dedicated headers). Further, we provide more details about the data augmentation.
>
> **“The “Heatmap regression configuration” and “uncertainty quantification” sections are not clear”**
>
> We agree that these sections might have been unclear in the initial manuscript and performed the following modifications:
> - The multimodality detection is separated from the first section
> - Each section now contains a clear content description at the beginning
>
> **“Table 1 is very unclear and confusing”**
>
> We updated the table and caption to improve clarity.
>
> **“Can the proposed method isolate data and model uncertainties?”**
>
> In our method, we explicitly model heteroscedastic data uncertainty (i.e., uncertainty that varies across inputs) as described in the refences literature (e.g., [1, 2]). We agree that the construction “model uncertainty” is misleading and removed the respective occurrences.
>
> **“Please indicate the limitations of the paper”**
>
> We added a dedicated limitations subsection (Sec. 4.5) and discuss the limitations regarding the implementation in clinical practice in Sec. 5.
>
> [1] 10.48550/arXiv.2203.09168
>
> [2] 10.48550/arXiv.2310.18953

---

> > ### Comment · Reviewer_xy9D · 2026-01-31
> >
> > I want to thank the authors for their consideration of my comments, rigors, and professionalism. They replied precisely to all of my questions and drastically improved the clarity and readability of the paper.
> >
> > I have a final request for the authors:
> > Could you clarify in the paper if the U-Net was pretrained, fine-tuned, or frozen during the training of your method? Or if the U-Net was end-to-end trained using their heteroscedastic loss? And what are the weights (networks or variables) updated by their heteroscedastic loss during training (it is not clear what is updated in an end-to-end fashion via their loss during training)?

---

> > > ### Author Response · Authors · 2026-01-31
> > >
> > > We sincerely thank the reviewer for the encouraging comments and are pleased that the revisions clarified the presentation and improved the overall readability of the paper.
> > >
> > > Regarding the follow-up question, the U-Net is neither pretrained nor frozen, but initialized with random weights and trained in an end-to-end fashion. All trainable parameters of the U-Net are optimized during training using gradients originating from the heteroscedastic loss (and the regularizer).
> > >
> > > To clarify the intuition: The U-Net outputs a spatial feature map $h$ to which a row-wise softmax is applied. In the resulting heatmap $\hat{h}$, each row with index $i$ can be interpreted as a pseudo-probability distribution, from which a mean $\mu_i$ and variance ${\sigma_i}^2$ are computed (Eq. 2). As per the method, the mean is defined as the prediction of the PM boundary position in this row, while the variance serves as a measure of uncertainty over this position. Both quantities are used in the heteroscedastic loss (Eqs. 3+4) together with the ground truth column index $y_i$ in row $i$ to optimize the U-Net.
> > > In this setup, during training, the U-Net is encouraged to produce distributions that are (1) well-localized around the true PM boundary, reflecting accurate predictions, and (2) appropriately wide where uncertainty is higher. In cases where the contour prediction is accurate, the second term of the loss dominates, encouraging the model to produce narrower distributions with lower variance, i.e., lower uncertainty. In cases where predicting the contour is difficult, e.g., due to occlusions or noise, the first term dominates, encouraging wider distributions with higher variance, i.e., increased uncertainty.
> > >
> > > We would be happy to get the opportunity for further clarification of this aspect in the manuscript for the camera-ready version.

---

### Official Review · Reviewer_CWUp · 2026-01-10

**Confidence:** 4
**Preliminary Rating:** 3
**Final Rating:** 4

**Summary:**

The authors propose a method for detection of pectoral muscle in mammography images. The main focus of the work is on the uncertainty of the predictions and their tolerance to the noise. The experiments are conducted in a publicly available datasets and several ablations are performed, in particular on the model complexity and the noise.

**Strengths:**

The authors address a relevant issue of the uncertainty of predictions in the context of pectoral muscle detection in mammography. The clinical context is introduced and the method motivation is given.

The paper is well structured and the method is clearly presented. The ablations are relevant and comprehensive.

**Weaknesses:**

Whlie the work is methodologically clear, the paper lacks some details with regard to the relevance of such appoach. That is, predicting row-wise column index is not within the state of the art and the more common methods are desigend to predict the segmentation masks. More discussion is needed to put light on the advantages of the proposed approach.

**Detailed Comments:**

The authors propose a method for pectoral muscle detection in medio-lateral oblique mammography views.

The authors model the task through a row-wise column-prediction and propose to asses the uncertainty throught the coordidante mean and standard deviation. It does not clarly stand out what is the advantage of such an approach and how the method compared to other uncertanty estimations. This prevents from understanding the interest of the proposed method. Could the authors discuss more the advantages?

The authors propose to model the dose reduction adding noise. This might not be clinically realistic. Could the authors provide more details about how the noise have been modeled and what parameters were used? Were any others perturbation techniques used, e.g., blur?

The authors mention multi-modality in the manuscript. This term might appear confusing as usually employed for different imaging modalities. Could the authors clarify?

**Justification Of Final Rating:**

The revised work address many of the concerns allowing for a better comprehension of the contribution and for a better reading flow through the paper. A more clear comparison to segmentation method and clarifying the pros and cons could be relevant. Nevertheless, the paper is still a worthy conference material.

**Justification Of The Preliminary Rating:**

While the paper is well structured, and the method is clearly presented, the advantages of the proposed method do not stand out from the manuscript. They shall be discussed further to consider acceptance.

**Questions To Address In The Rebuttal:**

See comments.

---

> ### Author Response · Authors · 2026-01-23
> **Rebuttal response to reviewer CWUp**
>
> We sincerely thank the reviewer for the thoughtful and valuable feedback. We address the points raised as follows:
>
> ---
>
>  **“The authors model the task through a row-wise column-prediction and propose to asses the uncertainty throught the coordidante mean and standard deviation. It does not clarly stand out what is the advantage of such an approach and how the method compared to other uncertanty estimations. This prevents from understanding the interest of the proposed method. Could the authors discuss more the advantages?”**
>
> We thank the reviewer for this comment. Motivated by anatomical prior knowledge, the boundary separating the pectoral muscle (PM) from the breast tissue is represented as a low-dimensional vector of row-wise column indices. This formulation was shown in Huemmer et al. (2024) to outperform standard pixel-wise classification baselines in terms of segmentation performance, parameter efficiency, and inference time. We agree that the connection to previous work and the advantages of this approach were not sufficiently clear in the initial manuscript. To address this, we have added a dedicated contributions section (Sec.2) to emphasize that our method advances Huemmer et al. (2024), and expanded the related work section (Sec. 2) on PM segmentation to more thoroughly discuss the benefits of column-index modeling.
>
> Regarding other uncertainty quantification methods: Previous work on uncertainty modeling in PM segmentation address mainly model uncertainty, i.e. epistemic uncertainty. Our approach aims to model heteroscedastic uncertainty to account for the discussed ambiguities within the input images in MLO mammograms. To the best of our knowledge, our study is the first approach to model heteroscedastic uncertainty in PM segmentation.  This is highlighted in the related work section on uncertainty quantification (Sec. 2). Furthermore, we agree that the relation to other heteroscedastic methods from other domains needed clarification. Therefore, we reframed and revisited the last paragraph of the related work section (Sec. 2) to discuss the relation of our method to these approaches. Specifically, we highlight the limitation of current heteroscedastic frameworks that model the row-wise error as a parametrization of a uni-modal Gaussian distribution and not by full probability distributions.
>
> ---
>
> **“The authors propose to model the dose reduction adding noise. This might not be clinically realistic. Could the authors provide more details about how the noise have been modeled and what parameters were used? Were any others perturbation techniques used, e.g., blur?”**
>
> We appreciate the concern regarding clinical realism. In this experiment, our primary aim was not to replicate clinically realistic dose reductions, but rather to assess whether a model trained to provide uncertainty estimates in the presence of input-dependent noise (occlusions etc.) can generalize to previously unsee sources of input-dependent noise. In other words, the focus is on testing the model’s ability to generalize to new sources of input-dependent noise, rather than simulating a clinically exact scenario. In the revised manuscript, we explicitly mention the generalizability aspect in the experiment description in Section 4.3.
>
> Nevertheless, this scenario might still be of clinical relevance for low-dose preshots as discussed in the Introduction (Sec. 1). These preshots are acquired at a significantly lower dose than full-dose mammograms which effectively results in notably increased Poisson noise.
>
> We extended the experimental description to clarify that Poisson noise is added to the unprocessed images as described in details in Eckert et al. (2019). To reproduce our setup only the effective dose reduction is required which is referred to as $\alpha$ in the range [0, 1] in the original publication. No further perturbations were applied.
>
> ---
>
> **“The authors mention multi-modality in the manuscript. This term might appear confusing as usually employed for different imaging modalities. Could the authors clarify?”**
>
> In our work, the terms “multimodality” and “multimodal” refer to properties of probability distributions, i.e., the presence of multiple modes. This usage is not uncommon in literature (see, e.g., [1]), but we agree that the term may be ambiguous due to the increasing focus on using different data types (e.g. imaging modalities) in recent scientific literature. We attempt to clarify this whenever possible by explicitly referring to multimodality as a property of a probability distribution in the revised version of the manuscipt.
>
>
> [1] Friedman, L., Hanson, T., & Komogortsev, O. V. (2021). Multimodality During Fixation - Part II: Evidence for Multimodality in Spatial Precision-Related Distributions and Impact on Precision Estimates. Journal of eye movement research, 14(3), 10.16910/jemr.14.3.4. https://doi.org/10.16910/jemr.14.3.4

---

> > ### Comment · Reviewer_CWUp · 2026-02-01
> >
> > I'd like to thank the authors for addressing my comments and concerns in a structured manner. The manuscript appears clearer now.
> >
> > The comparison to the segmentation methods is not yet perfectly clear for me. Specifically with Figure 2, that reports higher performances of smaller segmentation networks (that might be both superior in segmentation and inference time performance).
> >
> > Yet, this does not prevent me from voting for acceptance.

---

> > > ### Author Response · Authors · 2026-02-01
> > >
> > > We want to thank the reviewer once again for the constructive feedback and for acknowledging the improved clarity of the revised manuscript. We appreciate the reviewer’s additional comment and provide further clarification regarding the comparison of the segmentation performance (Sec 4.2 and Fig. 2) below.
> > >
> > > The aim of this experiment was to evaluate whether uncertainty modeling in the CI vector regression setup negatively affects segmentation performance. Therefore, we compare segmentation performance, against a traditional segmentation baseline: a U-Net performing pixel-wise classification to predict a binary mask.
> > >
> > > While our method achieves comparable performance for larger model configurations, a performance drop is observed for smaller models (depth <= 3 and filters <= 3). As discussed in Sec. 4.2, we attribute this to the increased task complexity: while the baseline U-Net only solves the segmentation task, the U-Net in out setup must learn to produce stable probability distributions to (1) solve the segmentation task, and (2) model uncertainty. For very small networks, the available model capacity appears insufficient for this joint objective. It is worth noting that the configuration at which the performance starts to drop noticeably (depth = 3, filters = 3) has fewer than 60k parameters.
> > >
> > > Regarding inference time, our method is based on the same U-Net architecture as the baseline and therefore has identical computational complexity in its main processing unit. The only additional operations are a softmax applied to the U-Net output and the computation of mean and variance in Eq. 2, which are implemented via simple summations and matrix multiplications. As a result, the added computational overhead is marginal, and the difference in inference speed is negligible, in particular for larger model configurations.

---

### Author Rebuttal · Authors · 2026-01-23

**Rebuttal:**

We thank all reviewers for their time and effort in reviewing our manuscript and for providing valuable and constructive feedback. In response, we have revised the paper to improve its structure, clarity, and readability. We have also incorporated the reviewers’ suggestions and concerns to the extent possible. All changes in the manuscript are highlighted in green. The revised manuscript is attached here.

We address the individual reviewers’ concerns in the corresponding comments

**Supporting Material:**

/attachment/aa0c618f26c5989ffd6cbf88e1cec9a94b11c678.pdf

---

### Meta-Review · Area_Chair_zKvc · 2026-02-03

**Recommendation:** Accept (Oral)
**Confidence:** 4

**Metareview:**

The [initial->final] reviewer scores are [borderline->weak accept] and 2x [weak accept -> strong accept], so all reviewers are in agreement about the paper's acceptance. In particular, the 2x strong accepts qualify this paper for an oral recommendation.

---

### Decision · Program_Chairs · 2026-02-13

Accept (Poster)